# Spectroscopic and Antibacterial Properties of CuONPs from Orange, Lemon and Tangerine Peel Extracts: Potential for Combating Bacterial Resistance

**DOI:** 10.3390/molecules26030586

**Published:** 2021-01-22

**Authors:** Pitso Tshireletso, Collins Njie Ateba, Omolola E. Fayemi

**Affiliations:** 1Food Security and Safety Niche Area, Faculty of Natural and Agricultural Sciences, North West University, Private Bag X2046, Mmabatho 2735, South Africa; pitsotshire@gmail.com (P.T.); Collins.Ateba@nwu.ac.za (C.N.A.); 2Department of Chemistry, Faculty of Natural and Agricultural Sciences, North West University, Private Bag X2046, Mmabatho 2735, South Africa

**Keywords:** green synthesis, citrus peels, copper oxide nanoparticles, antibacterial activity

## Abstract

Green synthesis of nanoparticles using citrus peel extracts is known to be environmentally friendly and non-toxic when compared to chemical methods. In this study, different citrus peel extracts obtained with the solvents acetone and distilled water were used to synthesize copper oxide nanoparticles (CuONPs). The nanoparticles were characterized using cyclic voltammetry, ultraviolet-visible spectroscopy, energy-dispersive X-ray spectroscopy (EDS), transmission electron microscopy (TEM), scanning electron microscopy (SEM) and Fourier-transform infrared spectroscopy (FTIR). The absorption spectrum of CuONPs prepared with acetone exhibited characteristic peaks at the wavelengths between 280–293 nm, while those with distilled water had peaks at 290 nm. The acetone-synthesized CuONPs were spherical while those produced using distilled water were rod-shaped. Based on EDS, the analysis revealed a trace spectrum of CuO nanoparticles with different weight compositions that varied with the type of citrus peel and solvent used. FTIR measurements were carried out in the range of 500–4000 cm^−1^ for citrus peel extract mediated CuONPs. The spectra had five vibrations occurring at approximately 473, 477, 482, 607 and 616 cm^−1^ for all samples, which can be attributed to the vibrations of CuO, validating the formation of highly pure CuONPs.

## 1. Introduction

According to the National Institute of Communicable Diseases (NICD), there are constant outbreaks and re-emergence of infections caused by pathogenic bacteria such as *Escherichia coli*, *Salmonella typhi* and *Klebsiella pneumonia* [1]. Infectious diseases cause suffering and death to humans and severe economic implications, which are not always immediately appreciated. Thus, the increase in the number of contagious diseases poses significant constraints to public health and healthcare facilities. Moreover, these microbes’ pathogenicity is also enhanced by the possession of antibiotic resistance determinants [2,3]. This is aggravated by the fact that antimicrobial resistance among bacterial pathogens is rising, making this an issue of severe public health concern [4]. Furthermore, due to the very little progress in the development of new antimicrobial agents, and given that the goal of all health-related measures mostly involves an effective treatment control strategy, sustainable investment in countering antimicrobial resistance needs to be ensured [5]. To achieve this, highly coordinated strategies that may facilitate the discovery of alternative antibacterial agents is imperative. Recently, nanoparticles have proven to be very effective against clinically relevant antibiotic-resistant pathogenic bacterial strains [6]. Much attention has been directed towards exploring the antimicrobial activity of various plant extracts and their components, and some have produced significant inhibitory effects against pathogenic bacteria, yeasts, fungi and viruses [7,8,9,10]. Among the potential candidates, citrus fruit extracts generate considerable interest due to the wide range of phytochemical components they possess [11,12].

Green synthesis is a set of principles that eliminates the use of hazardous substances in manufacturing and applications of chemical products [13]. There is a need for such methods since there are increasingly popular concerns associated with the environment and therefore we need “green” substances that are environmentally friendly [14]. The usage of plant extracts in green synthesis of nanoparticles is easy to perform and rapid, and production on the large scale is recently an emerging area of research [15]. This plays an essential role in nanotechnology, as using plant extracts is eco-friendly, cost-effective and is a better alternative over chemical methods [16]. Moreover, plant extracts are known to have a great significance in the treatment of microbial infections and have been used as a valuable source for maintaining human health [17].

Nanoparticles are tiny solid materials with a structural length of a few nanometers [18]. They are small and have a large surface to volume ratio; their size and shape-dependent properties are of interest in applications such as bio-sensing and optics [19]. There are many metal nanoparticle applications, mainly in pharmacy and medical sciences, especially in counteracting antimicrobial properties [20].

After the process of synthesis, it is necessary that nanoparticles are characterized primarily, to assess the safety properties especially if they are intended to be used at full potential to enhance human welfare (healthcare industry) in the area of nanomedicine [21]. Before the toxicity is assessed, parameters such as size distribution, shape, size, surface area, aggregation and solubility must be evaluated. Many analytical techniques have been used to characterize synthesized nanoparticles, including FTIR (Fourier-transform infrared spectroscopy), X-ray photoelectron spectroscopy (XPS), UV-Vis spectroscopy, XRD (X-ray diffraction), TEM (transmission electron microscopy) and SEM (scanning electron microscopy) [22].

Copper oxide is one of the metallic multifunctional nanoparticles with useful physical properties applied in biomedicine, electron chemical effects, antimicrobial studies, solar power generation and catalytic causes [23,24]. Copper oxide is a compound with a monoclinic structure used as a semiconductor [25]. Copper oxide nanoparticles have been used for applications such as antibacterials, in antifouling paint coatings to protect objects from microorganisms, water purifiers, fungicides and algaecides [26]. According to the United States Environmental Protection Agency, copper has been recognized as an antimicrobial material that displays enhanced antimicrobial activity towards pathogenic microorganisms [27]. Furthermore, the synthesis of copper oxide makes use of green chemistry methodology by making use of plant extracts [28].

El-Moslamy et al. [29] reported that CuONPs from citrus peels have phytochemicals that vary because they are found in different geographical regions, with orange peels being the richest in these compounds. Metal oxide nanoparticles from these citrus peels can anchor to bacterial cell walls and subsequently penetrate through them [30]. Additionally, they are capable of inhibiting cell synthesis of target bacteria [31,32]. A study by Vincent et al. [33] showed that synthesized CuONPs using *Citrus aurantifolia* leaves could be used as an antibacterial agent and photocatalyst for dye removal and purification of contaminated water in industries. Moreover, they reported that CuONPs from the same plant extract showed sensitivity against *K. pneumonia* and *S. aureus*, revealing that they have antibacterial properties capable of inhibiting drug-resistant bacteria. The tangerine plant extract has been used to remove inorganic and organic contaminants from wastewater [34]. Furthermore, nanoparticles produced from these plant extracts were evaluated as catalysts for methyl orange dye degradation, and the results confirmed that they could speed up the process of dye degradation [35].

One health challenge is resistance to treatments administered against various infections [36]. According to the NICD, there are constant outbreaks and re-emergence of pathologic complications caused by pathogenic strains such as *E. coli* O177, *E. coli* O157, *E. coli* O26, *Salmonella typhi* and *S. pneumonia* that are most often resistant to antibiotics [1]. The World Health Organization has documented an increase in the number of infectious diseases. It approved a global action plan on antimicrobial resistance aimed at ensuring sustainable investment in countering antimicrobial resistance. Therefore, there is an increasing need to produce safer and less environmentally harmful compounds that may be useful in antibacterial applications [37]. Antimicrobial agents are compounds that inhibit or slow down bacteria’s growth without being toxic to surrounding tissues; thus, the agents are also used to reduce multiple drug-resistant bacteria [26]. Nanoparticles of copper oxide have also been reported to exhibit antifungal and antibacterial properties, thus eliminating dust mites [38].

Comparative studies on citrus peel extracts used to synthesize metal oxide nanoparticles are limited in the literature. Therefore, the present study aimed to report a comparative study on spectroscopy, morphological, cyclic voltammetry and antimicrobial properties of CuO nanoparticles amalgamated from citrus peel extracts of orange, lemon and tangerine.

## 2. Results and Discussion

### 2.1. Synthesis of Nanoparticles

For this study, copper oxide nanoparticles were synthesized using citrus peels (orange, lemon and tangerine) with copper (II) nitrate (VI) pentahydrate ((CuNO_3_)_2_·5H_2_O) as a precursor. Acetone and distilled water were solvents. The color of the extracts was orange. The resultant precipitates were brownish-black for the acetone solvent and brown for the distilled water solvent. The extract and precursor solution changed from light blue to a green color solution, indicating a formation of copper oxide nanoparticles. A similar color change was observed in the literature [39].

### 2.2. UV-Vis

Quantitative determination to confirm and analyze the chemical structure of the nanoparticles with UV-Vis spectroscopy revealed absorption peaks at wavelengths ranging from 200–800 nm. Figure 1 shows the absorption spectrum of copper oxide nanoparticles synthesized from selected citrus peel extracts using distilled water as the solvent. Characteristic peaks were detected at a wavelength of 290 nm (Figure 1). The spectrum of copper oxide nanoparticles synthesized using acetone exhibited distinct peaks at wavelengths ranging from 285–293 nm (Figure 2). These findings align with previous reports in which the UV-Vis spectra for copper oxide nanoparticles ranged from 280–1000 nm [40,41].

Moreover, the UV-Vis spectra of copper oxide nanoparticles synthesized from Carica papaya ranged between 250–300 nm [42]. This was because confirmation of CuO nanoparticles was at an intense surface of plasmon resonance. To support another study [43], we also confirmed an absorbance peak at 285 nm with a UV-Vis spectrophotometer of the synthesized nanoparticles. In contrast, the literature shows a spectrum below the range reported [44], with absorption peaks between 230–280 nm for CuONPs. Moreover, another study [45] attributed to the surface plasmon resonance (SPR) of CuONPs a characteristic absorption peak of 275 nm. UV-Vis absorption spectra for metal oxide nanoparticles from different citrus peel extracts using acetone and distilled water as solvents showed a communal peak at 290 nm, indicating that they were close to the reported studies. Acetone solvent had the highest absorbance compared to the distilled water samples.

### 2.3. FTIR

The main principle of FTIR is that it helps us to identify functional groups present in the compound on the surface of the nanoparticles, by providing information of molecules and biomolecules present in the plant extracts used to prepare the metal oxide synthesis, which usually act as reducing and capping agents. Moreover, it is also used to determine the relative sizes and positions of all absorptions or peaks in the infrared regions. FTIR measurements shown in Figure 3 and Figure 4 were carried out by scanning in the range 500–4000 cm^−1^ for citrus peel extract mediated CuO nanoparticles. The characteristic peaks ranged from 600–3413 cm^−1^ and were more visible in the synthesized nanometal oxide. Several functional groups were sourced from the reducing agent which then appeared in metal oxide nanoparticles. The bands obtained at 1585 and 1644 cm^−1^ show the carbonyl C=O widening bonds. In the same way, the C–H stretching bonds occur in the region of 3300–2800 cm^−1^ [46]. Additionally, the broad band fixed at 3413 and 3392 cm^−1^ is attributed to the stretching and bending vibrations of absorbed water and surface hydroxyls [47]. The band shifting of the O–H groups in extract and nanoparticles around 3030 cm^−1^ constituted the involvement of phenols in the reduction of metal oxide [29]. Furthermore, medium peaks at 1367 and 2660 cm^−1^ were allocated to the C–H group. The bands obtained at 1585 and 1644 cm^−1^ show the carbonyl C=O broadening bonds. The vibration at 616 cm^−1^ can be attributed to the vibrations of Cu–O, confirming the formation of pure Cu–O metal nanoparticles [48]. FTIR spectra have five vibrations occurring at approximately 473, 477, 482, 607 and 616 cm^−1^ for all samples, which can be attributed to the vibrations of CuO, endorsing the formation of highly pure CuONPs.

### 2.4. Morphological Characterization

#### 2.4.1. SEM

SEM was used to confirm the morphology of nanoparticles synthesized. SEM images of CuONPs clearly show that the particles are rod and spherical, not forming any clusters shown in Figure 5. The type of solvent affected the nanoparticles’ morphology: using distilled water gave rod-like shapes while acetone yielded spherical shapes. Samat et al. [49] reported CuO nanoparticles at different concentrations and found that the particle size decreased when the CuO acetate was increased, giving a spherical morphology shape. Polat et al. [50] also reported that the micrographs were found to be smooth and rough, forming sphere-shaped NPs. Moreover, Al-Kalifawi [51] stated that the magnetic iron oxide from Hund Fruit (*Citrus medica*) extracts was round, which agrees with what was observed in this study. In contrast to the study of [52], the SEM images showed a rectangular morphology prepared from copper oxide nanoparticles. This type of shape was close to what was found in Figure 5, which shows a picture of clustered rods. Figure 6 shows that the nanoparticles are spherical and have formed agglomerated particles; these nanoparticles were obtained from the CuONPs with acetone solvent.

#### 2.4.2. EDS (Energy-Dispersive X-ray Spectroscopy)

EDS (energy-dispersive X-ray spectroscopy) was used to confirm the presence of both copper and oxygen in the nanoparticles. CuONPs had different weight compositions varying with the type of citrus peel and solvent used. The weight composition spectra of copper and oxygen for the distilled water solvent shown in Table 1 were as follows: lemon had 81.73% and 17.33%, tangerine had 75.49% and 24.51%, and orange had 69.52% and 30.48%, respectively. The acetone solvent spectra were as follows: lemon extract had 72.62% and 27.38%, tangerine had 66.7% and 32.83% and orange had 39.04% and 45.27%, respectively. The lemon extract nanoparticles with distilled water solvent showed the highest percentage of copper composition, the reason for having higher antimicrobial activity.

#### 2.4.3. TEM

The CuO nanoparticle size and morphology were investigated with TEM images presented in Figure 7 and Figure 8 at different magnifications. Ren et al. [25] reported that the NNPs were nearly spherical and had an average CuO diameter that ranged between 22.4–94.8 nm. One study [53] gave images of dispersion of CuONPs with an average size of 44.5 nm. Moreover, as in Ref. [54], most of the CuONPs were spherical and had a mean of 52.51 nm. Figure 7 and Figure 8 show TEM images, which on close observation showed globular shapes; however, the size estimation and the subsequent presentation in different histograms revealed an average size ranging between 48–76 nm. These are more similar to previous reports. While Figure 7 revealed that the CuONPs prepared using acetone as solvent had no frequency distribution due to its irregular shapes, Figure 8a showed that the CuONPs-ODI had a frequency distribution of 74 nm, Figure 8b CuONPs-LDI had a frequency distribution of 50 and Figure 8c CuONPs-TDI had a frequency distribution of 70 nm. The nanoparticles CuONPs-ODI, CuONPs-LDI and CuONPs-TDI were obtained from the solvent of distilled water.

#### 2.4.4. Antimicrobial Study

The antibacterial efficiency of the citrus peel mediated CuONPs was evaluated against both Gram-negative (*C. coli* ATCC 33559, *E. coli* ATCC 25922, *M. catarrhalis* ATCC 25240, *S. diarizonae* ATCC 12325, *P. aeruginosa* ATCC 27853) and Gram-positive (*C. perfringens* ATCC 13124, *S. aureus* ATCC 25923, *L. monocytogenes* ATCC 19115, *S. pneumonia* ATCC 13883, *E. faecalis* ATCC 29212) bacteria using the agar well diffusion method. The results of the agar well diffusion assays indicating the antibacterial activity of CuO-NPs-O on pathogenic bacterial strains are summarized in Table 2, Table 3 and Table 4. The highest antibacterial activity was obtained with CuONPs synthesized from orange and lemon peels using acetone as a solvent since they exhibited antibacterial efficiencies against most organisms used in the study (Table 2 and Figure 9). These acetone-produced CuONPs were very active against *C. perfringens* (ATCC 13124), *E. coli* (ATCC 25922), *S. aureus* (ATCC 25923), *L. monocytogenes* (ATCC 19115), *S. pneumonia* (ATCC 13883), *P. aeroginosa* (ATCC 27853) and *M. catarrhalis* (ATCC 25240). Table 2 shows that CuO-NPs-OA (25 µg/mL and 50 µg/mL) exhibited activity against *C. perfringens* ATCC 13124 (12 mm and 19 mm), *E. coli* ATCC 25922 (18 mm and 24 mm), *M. catarrhalis* ATCC 25240 (7 mm and 16 mm), *C. coli* ATCC 33559 (20 mm and 26 mm) and *S. aureus* ATCC 25923 (13 mm and 25 mm) while CuO-NPs-ODI showed no activity. From data in Table 3, aqueous tangerine extract mediated CuO-NPs with concentrations 25 µg/mL and 50 µg/mL showed that CuO-NPs-TDI had activity only against *S. pneumonia* with a ZBGID of 8 mm and 14 mm, respectively, while those from acetone showed no activity against the pathogens. Aqueous lemon peel mediated CuO-NPs (CuO-NPs-LDI) showed activity against *C. coli* ATCC 33559, with a ZBGID of 16 mm at 25 µg/mL and 25 mm when 50 µg/mL of the nanoparticle was used (Table 4). Antibacterial efficiency was also exhibited against the Gram positive bacteria (*S. aureus*) with a ZBGID of 17 mm and 23 mm for concentrations 25 µg/mL and 50 µg/mL, respectively. CuO-NPs-LDI produced ZBGID of 9 mm and 13 mm against *L. monocytogenes* while those for the Gram negative *P. aeruginosa* were 6 mm and 20. Acetone lemon peel (CuO-NPs-LA) mediated CuONPs on the other hand showed no antibacterial activity. Generally, the antibacterial properties of the CuONPs increased with increase in concentration, as revealed by an increase in the respective ZBGIDs. In a previous study, copper oxide nanoparticles showed mild inhibitory effects against *E. faecalis*, *Salmonella*, *E. coli* and *S. aureus* [55]. Moreover, the findings of another study revealed that CuONPs displayed high antibacterial activity against two different bacteria strains, *viz E. aerogenes* (Gram negative) and *S. aureus* (Gram positive) [56]. A study by Ahamed et al. [57] also revealed that copper oxide nanoparticles were active against various bacterial strains that comprised *E. coli*, *P. aeruginosa*, *K. pneumonia*, *E. faecalis*, *S. flexneri*, *S. typhimurium*, *P. vulgaris* and *S. aureus*. Similar to these findings, a previous report [16] also revealed that the ZBGID increased when a concentration of aloe vera mediated CuONPs was used in the treatment of bacterial pathogens from fish. Our findings also revealed that the most effective CuO-mediated nanoparticles were from CuONPs-LDI.

## 3. Materials and Methods

The citrus peels were obtained from citrus fruit bought at Food Lover’s Market in Mafikeng, North-West South Africa. Reagents used included copper (II) nitrate (VI) pentahydrate ((CuNO_3_)_2_·5H_2_O; GlassWorld, Johannesburg, South Africa), sodium hydroxide (NaOH; Promark Chemicals, Johannesburg, South Africa), acetone (a product of VWR chemicals, Fontenay-sous-Bois, France) and distilled water. All reagents were of analytical grade. The pathogenic bacterial strains were obtained from the American type culture collection strains purchased from Bio Mérieux, South Africa.

### 3.1. Preparation of Citrus Peel Extract

Citrus peels of lemon, tangerine and orange were bought and washed with tap and deionized water several times. The peels were left for three days in the sun to dry and were further ground into powder using a blender. The lemon extract was prepared by weighing about 4 g of powder into 150 mL of deionized water, followed by 20 min of magnetic stirring and boiling. The extract was then filtered and stored at 4 °C for further analysis. The same procedure was used for orange and tangerine.

### 3.2. Synthesis of Copper Oxide Nanoparticles CuONPs

CuONPs were synthesized with slight modifications following the protocol described by [39]. A 200 mL solution of (CuNO_3_)_2_·5H_2_O was treated with an extract of 20 mL and was magnetically stirred at room temperature for 30 min until the light blue color changed to a green color, which indicated the formation of CuONPs. The solution was stirred for 1 h, then heated at 80 °C for 10 min. Then, 5 mL of 1 M sodium hydroxide (reducing agent) was added twice at 5 min intervals. The solution resulted in a brown precipitate that was repeatedly washed with deionized water followed by ethanol to remove impurities found in the precipitate. Lastly, it was dried at 60 °C and put in storage for analysis.

### 3.3. Characterization of the Synthesized Metal Oxide Nanoparticles

#### 3.3.1. Spectroscopy Characterization

The resulting synthesized metal oxide nanoparticles were characterized by energy-dispersive X-ray spectroscopy (EDS), UV-Visible spectroscopy, scanning electron microscope (SEM), transmission electron microscopy (TEM) and Fourier-transform infrared spectroscopy (FTIR). UV-Vis spectroscopy is a simple, fast, sensitive and selective characterization technique that enables the determination of nanoparticle formation and stability [39]. The metal oxide nanoparticles (copper oxide) synthesized from lemon, tangerine and orange using solvents acetone and distilled water (2 µg) were weighed out and distilled water was added then sonicated. The resulting solutions were poured into cuvettes and spectra in the range of 200–800 nm were determined using the UV-Visible spectrometer. FTIR spectra of the powdered samples of the synthesized metal oxide nanoparticles were measured using the Cary 600 series FTIR spectrometer, Bridgewater, NJ, USA. FTIR analysis was carried out to identify the surface functional groups and detect the stretching and bending vibrations in the lemon, orange and tangerine extracts and nanoparticle samples. For measurements of FTIR, a small amount of powdered NPs was placed on round-shaped selenide plates for spectral analysis.

#### 3.3.2. Morphological Characterization

Morphology, microstructure and the elemental composition of the synthesized metal oxide nanoparticles were examined using SEM with EDS. Scanning electron microscopy (SEM) images were recorded using a Hitachi 3600 SEM instrument (Shizuoka, Japan) and energy dispersive X-ray analysis (EDS) was done using the Thermo Fisher Scientific Ultradry (Madison, WI, USA) instrument. TEM analysis aims to provide information on the shape and size distribution of synthesized nanoparticles while confirming metal oxide nanoparticle existence in the synthesized samples. TEM analysis was done using the transmission electron microscope, JEOLJEM 2100 electron microscope (Tokyo, Japan) operated at 200 kv accelerating voltage and connected to an energy-dispersive spectrophotometer (EDS). In performing the analysis, 2 mg of each sample was weighed out into a sample bottle and about 4 mL ethanol solution was added. The mixture was sonicated for 10 min to disperse the particles, using a digital ultrasonic cleaner. A drop of the sonicated mixture was placed on a carbon-coated copper grid and allowed to dry for about 5 min.

#### 3.3.3. Antibacterial Activity of the Synthesized CuONPs

The antibacterial potency of the synthesized metal oxide nanoparticles was evaluated using ten pathogenic bacterial strains associated with both food poisoning diseases and nosocomial infections in humans. Each bacterial strain was sub-cultured in the nutrient broth and incubated aerobically at 37 °C for 24 h. (CuNO_3_)_2_·5H_2_O [58], as a precursor to synthesize copper oxide nanoparticles, citrus peels (lemon, orange and tangerine), five strains of Gram-positive (*E. faecalis* ATCC 29212, *S. aureus* ATCC 25923, *L. monocytogenes* ATCC 19115, *S. pneumonia* ATCC 13883 and *C. perfringens* ATCC 13124) and five strains of Gram-negative (*E. coli* ATCC 25922, *M. catarrhalis* ATCC 25240, *S. diarizonae* ATCC 12325, *C. coli* ATCC 33559 and *P. aeruginosa* ATCC 27853) bacteria were used in the analysis. This study’s pathogenic bacterial strains were American type culture collection strains purchased from Bio-Mérieux, South Africa. Wells were made using sterile cork borer (10 mm) into each MHA Petri plate and were used to evaluate the antimicrobial activity of the synthesized CuO nanoparticles. The microbial culture was exposed to CuONPs of different concentrations (25 and 50 mg/mL). The zones of inhibition were observed and measured. DMSO (dimethyl sulfoxide) was used as negative control and tetracycline used as a positive control. The bacteria were cultured overnight on MHA at 37 °C in an incubator.

## 4. Conclusions

This study reports on the green synthesis of CuONPs prepared by the biosynthesis technique using citrus extracts. The UV-Vis absorption spectra show that copper oxide nanoparticles synthesized from selected peel extracts exhibit characteristic peaks at the wavelengths 280–320 nm. SEM images reveal that the particles appear to be almost rod-like and sphere-shaped. TEM images showed spherical shapes and the size estimation, and the subsequent presentation in different histograms revealed an average size ranging between 48–76 nm. The CuO nanoparticles exhibited significant antibacterial activity against *C. perfringens* ATCC 13124, *C. coli* ATCC 33559, *E. coli* ATCC 25922, *L. monocytogenes* ATCC 19115, *S. pneumonia* ATCC 13883, *P. aeroginosa* ATCC 27853 and *M. catarrhalis* ATCC 25240. *S. aureus* ATCC 25923 and *E. coli* ATCC 25922 showed a significant zone of inhibition to CuONPs compared to the positive control (tetracycline). *C. perfringens* ATCC 13124, *L. monocytogenes* ATCC 19115, *S. pneumonia* ATCC 13883, *P. aeroginosa* ATCC 27853 and *M. catarrhalis* ATCC 25240 strains showed a moderate zone of inhibition compared to the positive control (tetracycline). This study successfully validates the use of copper oxide nanoparticles with citrus extracts to exert antimicrobial properties on various microorganisms that are resistant to antibiotics.

## Figures and Tables

**Figure 1 molecules-26-00586-f001:**
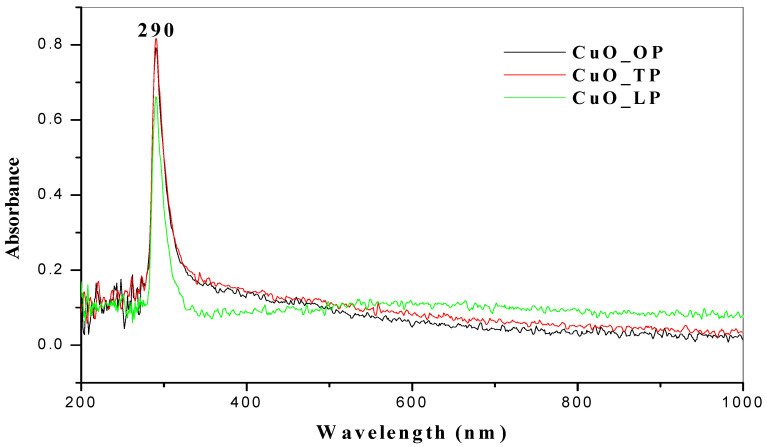
UV-Vis spectra of CuONPs from citrus peels extracted with distilled water solvent.

**Figure 2 molecules-26-00586-f002:**
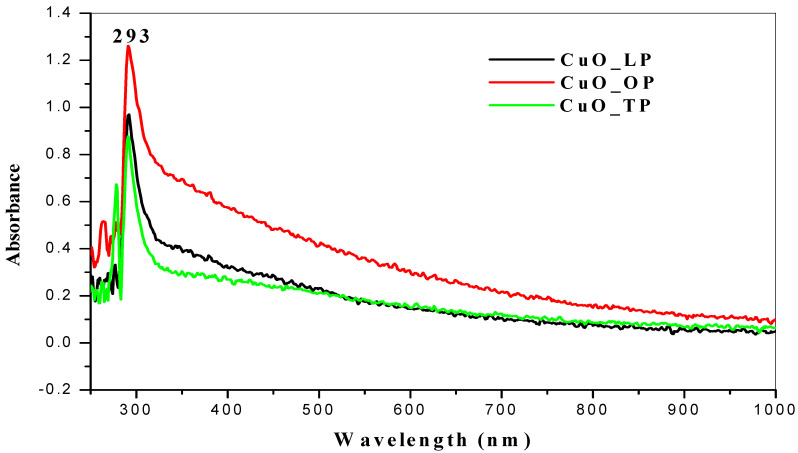
UV-Vis spectra of CuONPs from citrus peels extracted with acetone solvent.

**Figure 3 molecules-26-00586-f003:**
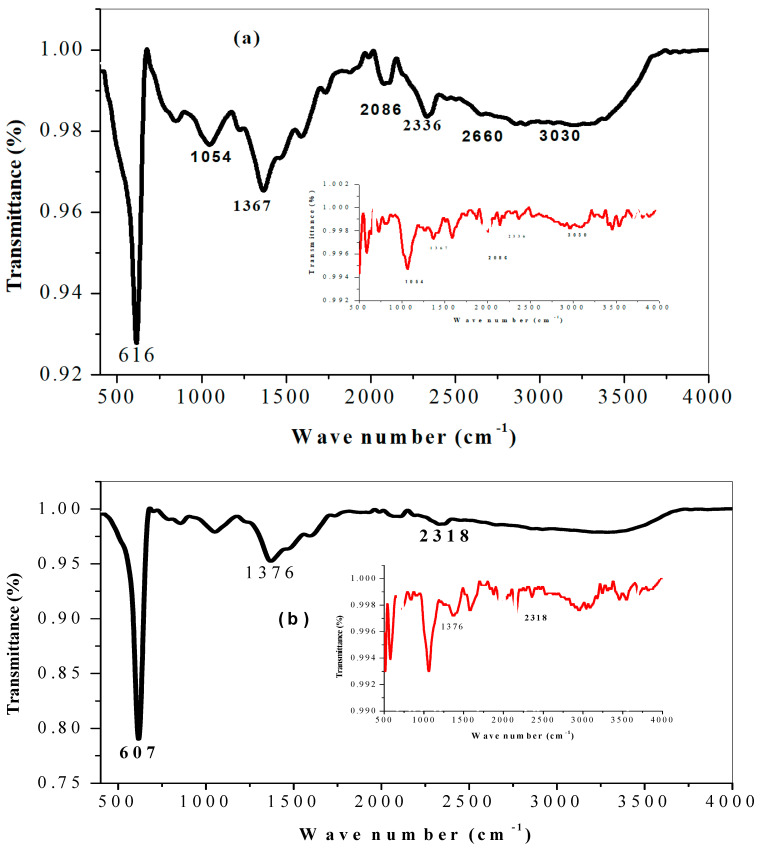
The FTIR spectra of CuONPs from (**a**) orange (**b**) tangerine and (**c**) lemon peel extracts in distilled water; insert is the spectra for the respective citrus extracts.

**Figure 4 molecules-26-00586-f004:**
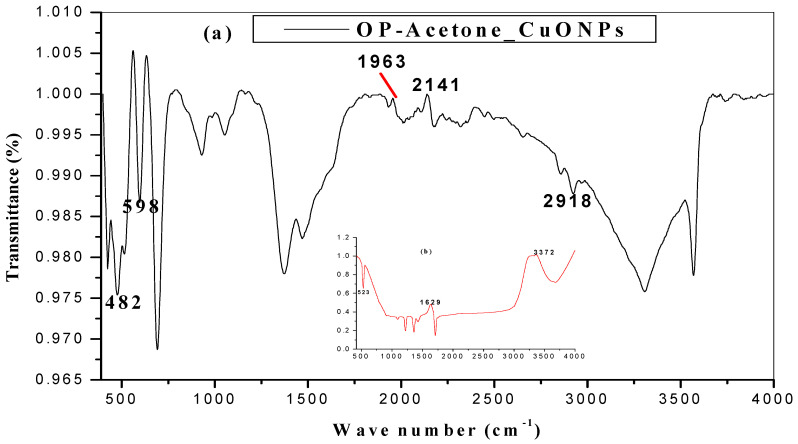
The FTIR spectra of CuONPs from (**a**) orange (**b**) lemon and (**c**) tangerine peel extracts in acetone; insert is the spectra for the respective citrus extracts.

**Figure 5 molecules-26-00586-f005:**
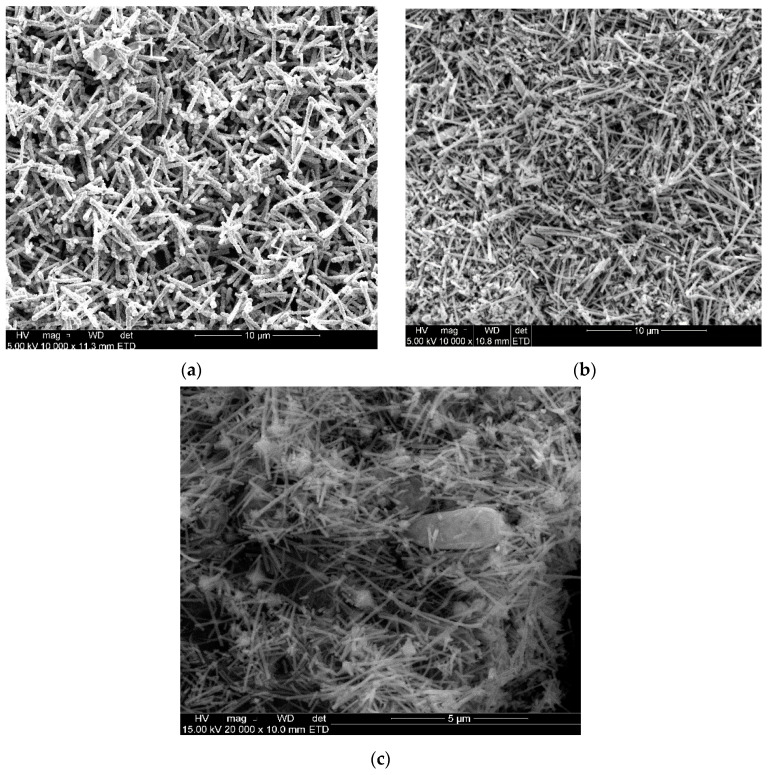
SEM images of CuONPs from citrus peel extracts using distilled water as a solvent. (**a**) Lemon, (**b**) tangerine and (**c**) orange.

**Figure 6 molecules-26-00586-f006:**
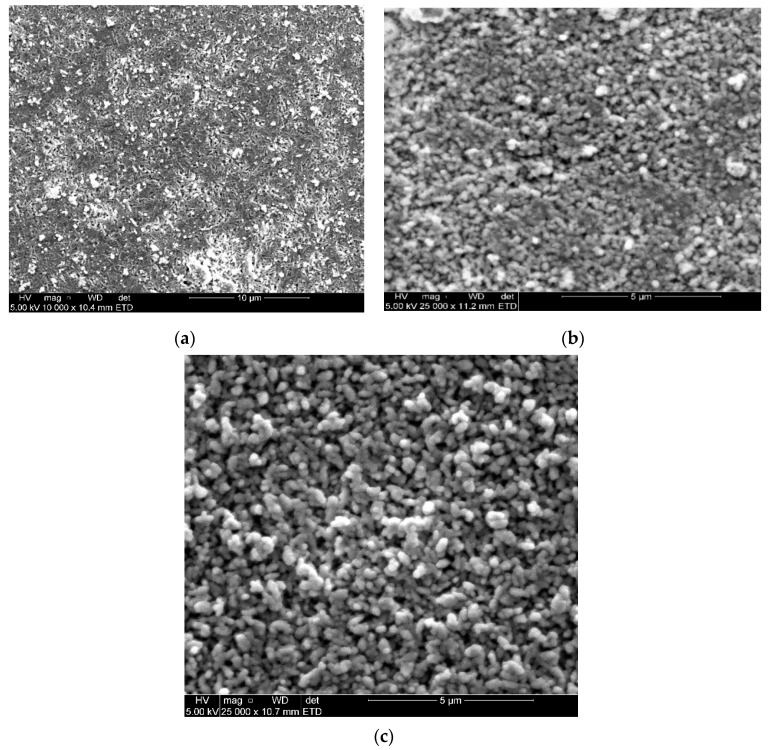
SEM images of CuONPs from citrus peel extracts using acetone as a solvent. (**a**) Lemon, (**b**) tangerine and (**c**) Orange.

**Figure 7 molecules-26-00586-f007:**
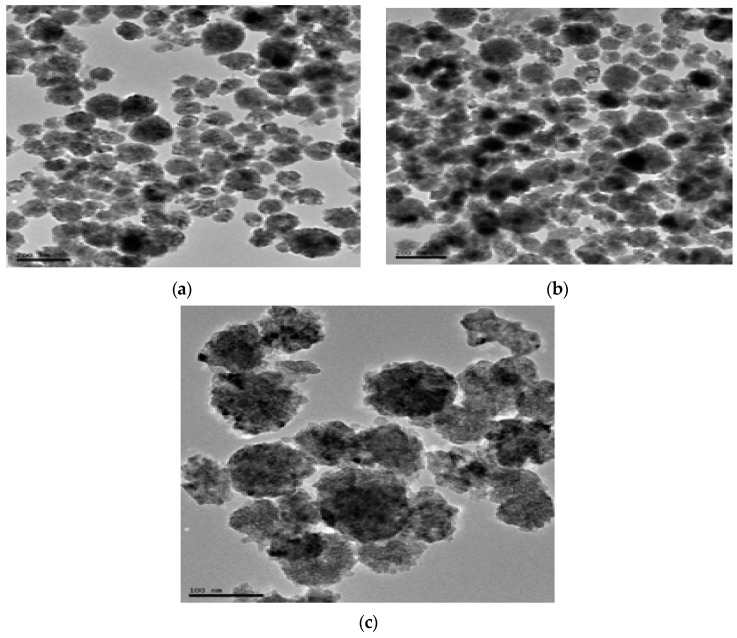
TEM (transmission electron spectroscopy) images for (**a**) CuO-NPs-LA, (**b**) CuO-NPs-TA, and (**c**) CuO-NPs-OA from lemon, tangerine and orange extracts, respectively, using acetone as solvent.

**Figure 8 molecules-26-00586-f008:**
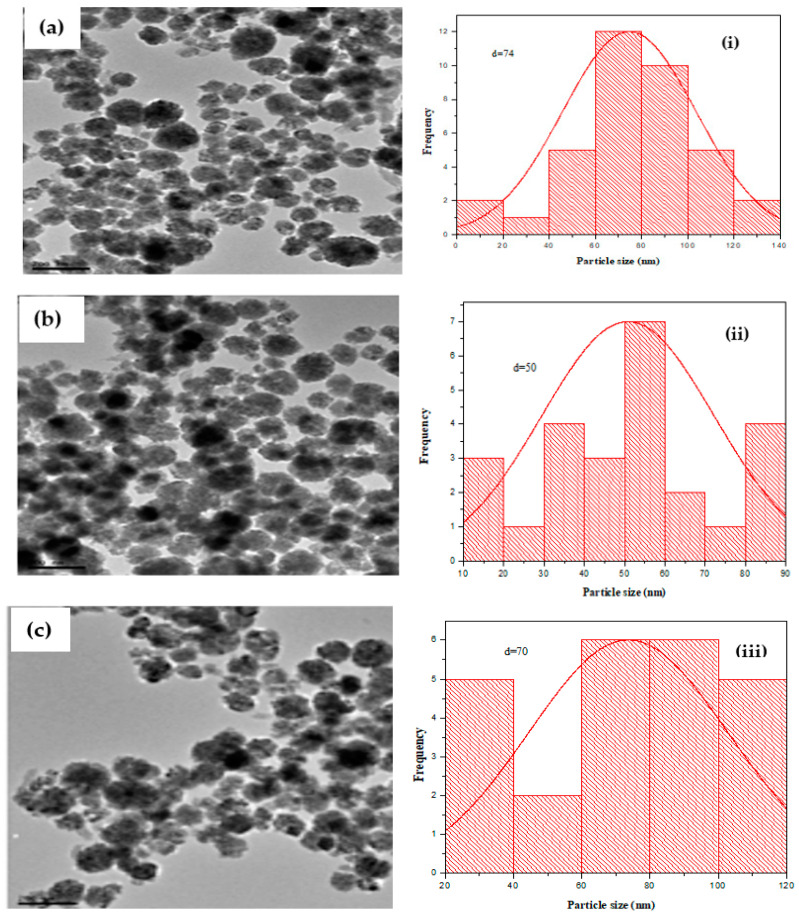
TEM images for (**a**) CuO-NPs-ODI, (**b**) CuO-NPs-LDI and (**c**) CuO-NPs-TDI from orange, lemon and tangerine extracts, respectively, using distilled water as solvent and their size distribution histogram. (**i**) CuO-NPs-ODI, (**ii**) CuO-NPs-LDI and (**iii**) CuO-NPs-TDI.

**Figure 9 molecules-26-00586-f009:**
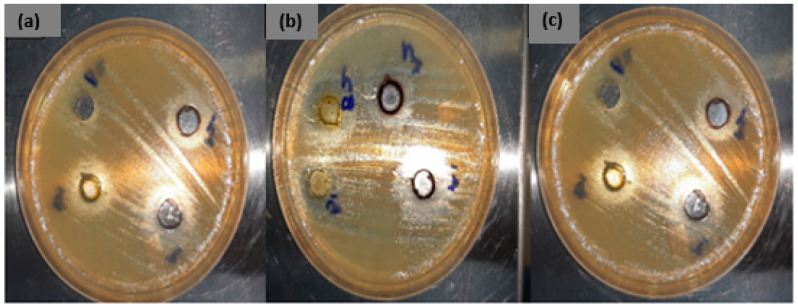
Antimicrobial sensitivity test of copper oxide nanoparticles. (**a**) *Escherichia coli*, (**b**) *Campylobacter coli* and (**c**) *Staphylococcus aureus.*

**Table 1 molecules-26-00586-t001:** EDS (energy-dispersive X-ray spectroscopy) data for CuONPs from citrus extracts of lemon, orange and tangerine using water and acetone solvents.

CuONPs from Citrus Extract/Solvent	% Cu	% O
Lemon/water	81.73	17.33
Orange/water	69.52	30.48
Tangerine/water	75.49	24.51
Lemon/acetone	72.62	27.38
Orange/acetone	39.04	45.27
Tangerine/acetone	66.70	32.83

**Table 2 molecules-26-00586-t002:** Antibacterial activity of CuO-NPs-O on pathogenic bacterial strains.

Pathogenic Bacterial Strains	Influence of Different Concentrations of CuONPs (Orange Extract) on Zone of Inhibition			
	(25 µg/mL) (Acetone)	(25 µg/mL) (Deionized Water)	(50 µg/mL) (Acetone)	(50 µg/mL) (Deionized Water)
*C. perfringens*	12 mm	0 mm	19 mm	0 mm
*C. coli*	20 mm	0 mm	26 mm	0 mm
*E. coli*	18 mm	0 mm	24 mm	0 mm
*S. aureus*	13 mm	0 mm	25 mm	0 mm
*L. monocytogenes*	0 mm	0 mm	0 mm	0 mm
*S. pneumonia*	0 mm	0 mm	0 mm	0 mm
*P. aeruginosa*	0 mm	0 mm	0 mm	0 mm
*M. catarrhalis*	7 mm	0 mm	16 mm	0 mm
*S. diarizonae*	0 mm	0 mm	0 mm	0 mm
*E. faecalis*	0 mm	0 mm	0 mm	0 mm

**Table 3 molecules-26-00586-t003:** Antibacterial activity of CuO-NPs-T on pathogenic bacterial strains.

Pathogenic Bacterial Strains	Influence of Different Concentrations of CuONPs (Tangerine Extract) on Zone of Inhibition			
	(25 µg/mL) (Acetone)	(25 µg/mL) (Deionized Water)	(50 µg/mL) (Acetone)	(50 µg/mL) (Deionized Water)
*C. perfringens*	0 mm	0 mm	0 mm	0 mm
*C. coli*	0 mm	0 mm	0 mm	0 mm
*E. coli*	0 mm	0 mm	0 mm	0 mm
*S. aureus*	0 mm	0 mm	0 mm	0 mm
*L. monocytogenes*	0 mm	0 mm	0 mm	0 mm
*S. pneumonia*	0 mm	8 mm	0 mm	14 mm
*P. aeroginosa*	0 mm	0 mm	0 mm	0 mm
*M. catarrhalis*	0 mm	0 mm	0 mm	0 mm
*S. diarizonae*	0 mm	0 mm	0 mm	0 mm
*E. faecalis*	0 mm	0 mm	0 mm	0 mm

**Table 4 molecules-26-00586-t004:** Antibacterial activity of CuO-NPs-L on pathogenic bacterial strains.

Pathogenic Bacterial Strains	Influence of Different Concentrations of CuONPs (Lemon Extract) on Zone of Inhibition			
	(25 µg/mL) (Acetone)	(25 µg/mL) (Deionized Water)	(50 µg/mL) (Acetone)	(50 µg/mL) (Deionized water)
*C. perfringens*	0 mm	0 mm	0 mm	0 mm
*C. coli*	0 mm	16 mm	0 mm	25 mm
*E. coli*	0 mm	0 mm	0 mm	0 mm
*S. aureus*	0 mm	17 mm	0 mm	23 mm
*L. monocytogenes*	0 mm	9 mm	0 mm	13 mm
*S. pneumonia*	0 mm	0 mm	0 mm	0 mm
*P. aeruginosa*	0 mm	6 mm	0 mm	10 mm
*M. catarrhalis*	0 mm	0 mm	0 mm	0 mm
*S. diarizonae*	0 mm	0 mm	0 mm	0 mm
*E. faecalis*	0 mm	0 mm	0 mm	0 mm

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
