# Peer review of "Spectroscopic and Antibacterial Properties of CuONPs from Orange, Lemon and Tangerine Peel Extracts: Potential for Combating Bacterial Resistance"

_molecules, 2021, doi:10.3390/molecules26030586_

Round 1
Reviewer 1 Report
the title needs to be change, it use Cu(NO3)2 to synthesize not orange, lemon and tangerine peel extracts. peel extracts only play a role as acids I think.
there are many big flaws.
Introduction needs to show some research about range, lemon and tangerine peel extracts for synthesize CuO NPs, what mechanism??
this is not related to this paper:"
"Citrus is a genus of flowering trees and shrubs found in the family of Rutaceae, which include
88 approximately 160 genera and 1700 species, and are globally used in herbal medicine [21]. They are
89 known to be the foremost source of dietary flavonoids, which are rare on other plants; as well as rich
90 in vitamins, minerals and their antioxidant properties. Flavonoids have antioxidant properties that
91 work directly against free radical scavengers within the body, which helps inhibit pathogenic
92 diseases and cell proliferation [22]. They play a protective role by inhibiting the invading pathogens
93 such as fungi, bacteria, and viruses [23].
94 Orange peels are popularly used to treat central nervous system disorders; they are also known
95 to produce protective enzymes that stimulate the immune system, block any damage on the genetic
Molecules 2020, 25, x FOR PEER REVIEW 3 of 22
96 material, protect human from cardiovascular diseases and lastly against cancer by use of specific
97 phytonutrients called flavonoids [24]. Orange and lemon plants have been proved one of the most
98 effective antifungal components [25]. Besides, citrus seeds are used in separating oils for plastics,
99 soaps and cooking oil. The essential oil contains aromatic compounds, terpenes and oxygenated
100 derivatives [26]. They bring into being inhibitory on gram-negative and gram-positive microbes; and
101 are used in food industries as natural antimicrobials [27]"
IT should be deleted!!!!
FTIR only can see characteristics but not quantities. It is better to see XPS. do XPS ?
Also SEM data is not clear and should be deleted. and all EDS data can merge to a table.
for antimicrobial assay, what is MIC and MAC dose?
you need to do MIC and MBC experiments.
Author Response
Response to Reviewer 1
|
Comments |
Authors Response
|
|
the title needs to be change, it use Cu(NO3)2 to synthesize not orange, lemon and tangerine peel extracts. peel extracts only play a role as acids I think |
Thanks for your suggestion. Note sir, the citrus peel extracts were used as solvent in the synthesis of CuO nanoparticles and they also served as capping agent.
|
|
there are many big flaws. Introduction needs to show some research about range, lemon and tangerine peel extracts for synthesize CuO NPs, what mechanism??
|
Thanks. New information with regards to research about range, lemon and tangerine peel extracts for synthesize CuO NPs, has been added on lines 79-90, page 2.
|
|
this is not related to this paper:" "Citrus is a genus of flowering trees and shrubs found in the family of Rutaceae, which include IT should be deleted!!!!
|
Thanks session deleted and new detail relating to the study added. |
|
FTIR only can see characteristics but not quantities. It is better to see XPS. do XPS ?
|
Thanks your suggestion is well noted. Unfortunately, we cannot embark on XPS analysis because of Covid restrictions as the equipment is not in our campus, but we shall consider doing the characterization and showing it in our future studies.
|
|
Also SEM data is not clear and should be deleted. and all EDS data can merge to a table.
|
Thanks. SEM images were changed to a better quality and the EDS data merged in a table. See lines 181 – 202 of the manuscripts.
|
|
For antimicrobial assay, what is MIC and MAC dose?
|
Minimum inhibition concentration and Minimum Alveolar Concentration |
|
you need to do MIC and MBC experiments.
|
Thank you so much for the recommendation. Due to Covid-19 restrictions and closing of the University I cannot carry out more experiments.
|
Reviewer 2 Report
Title: Spectroscopic and antibacterial properties of CuONPs from orange, lemon and tangerine peel extracts: potential for combating bacterial resistance
In this manuscript, the authors investigated green synthesis of nanoparticles using different plant extracts. Moreover, they used citrus peel, which is considering as a waste. Therefore, the topic of study deserves to be investigated. However, a lot of works on similar topic has already been published and I cannot recommend publishing of this manuscript in this form.
Some of the recommendations are as follows:
Abstract is too long. As I saw in Instructions for Authors, the abstract should contain 200 words. Please adjust the abstract and include just the most important findings. Also, exclude the comprehensive introduction at the beginning of the abstract.
Please include in the Introduction section previously published studies on the same or similar topic. Also, one manuscript on a very similar topic, titled “Nanotechnology bioprocess strategies and characterization of anti-multidrug resistant human pathogens copper/copper oxide nanoparticles from citrus peel waste extracts“ has already been published. Therefore, please include this study in the Introduction and explain what additional innovative findings and contributions are provided by the present study.
Full name and abbreviation NICD are mentioned twice, in Rows and 55 and 119. After the first use, please continue with using the abbreviation or use full name in both cases without abbreviation.
Rows 96-97 “specific phytonutrients called flavonoids” flavonoids are already mentioned before. Therefore, rephrase this and delete “specific phytonutrients called”
Row 127 Please, instead of “kill dust mites“ use more formal language. Some suggestions: eliminate, remove, etc.
Section “Preparation of citrus peel extract” should be revised and include detailed data regarding: particle size and moisture content of used citrus peels, exact way of stirring and producer of the magnetic stirrer, producer of the blender, and explanation of the drying procedure.
Additionally, include the producer of all used materials: acetone, copper nitrate, sodium hydroxide, etc.
Extraction procedure is ”4 g of powder into 150 ml of deionized water, followed by 20 min of stirring and boiling“. Please explain why you used this procedure and this ratio of solid/liquid.
Also, having in mind the boiling step, it is possible that most of the significant bioactive compounds are destroyed due to degradation. Therefore, it is necessary to optimize this extraction procedure and chemically characterize obtained extracts.
It is stated that the synthesis of copper oxide nanoparticles was conducted according to reference 62. Since it is well-known that some parameters such as extract volume, concentration, time, pp,mmtemperaturepH value, temperature, etc. can have an influence on the synthesis of nanoparticles, my recommendation is to include an explanation of influence of some parameters and explain why you chose the specific parameters. Also, pH value should be included.
Regarding antibacterial activity, it would be an improvement if you include an antibiotic as a reference.
Author Response
Response to Reviewer 2
Title: Spectroscopic and antibacterial properties of CuONPs from orange, lemon and tangerine peel extracts: potential for combating bacterial resistance
|
Comments |
Response
|
|
In this manuscript, the authors investigated green synthesis of nanoparticles using different plant extracts. Moreover, they used citrus peel, which is considering as a waste. Therefore, the topic of study deserves to be investigated. However, a lot of works on similar topic has already been published and I cannot recommend publishing of this manuscript in this form. Some of the recommendations are as follows: Abstract is too long. As I saw in Instructions for Authors, the abstract should contain 200 words. Please adjust the abstract and include just the most important findings. Also, exclude the comprehensive introduction at the beginning of the abstract.
|
Thanks, the abstract now has less than 200 words, the introduction session has also been revised. See yellow highlight on page 1 – 3 of the manuscript.
|
|
Please include in the Introduction section previously published studies on the same or similar topic. Also, one manuscript on a very similar topic, titled “Nanotechnology bioprocess strategies and characterization of anti-multidrug resistant human pathogens copper/copper oxide nanoparticles from citrus peel waste extracts“ has already been published. Therefore, please include this study in the Introduction and explain what additional innovative findings and contributions are provided by the present study.
|
Thank you so much for the comment. I went through the paper and added innovative findings. Please see lines 93-94. |
|
Full name and abbreviation NICD are mentioned twice, in Rows and 55 and 119. After the first use, please continue with using the abbreviation or use full name in both cases without abbreviation.
|
Thanks for your comment. The second full name of NICD has been removed.
|
|
Rows 96-97 “specific phytonutrients called flavonoids” flavonoids are already mentioned before. Therefore, rephrase this and delete “specific phytonutrients called”
|
Thanks, the paragraph was rephrased |
|
Row 127 Please, instead of “kill dust mites“ use more formal language. Some suggestions: eliminate, remove, etc |
Thanks, the word was replaced with eliminate. See line 106
|
|
Section “Preparation of citrus peel extract” should be revised and include detailed data regarding: particle size and moisture content of used citrus peels, exact way of stirring and producer of the magnetic stirrer, producer of the blender, and explanation of the drying procedure.
|
Thanks. The section was revised and the explanation was added
|
|
Additionally, include the producer of all used materials: acetone, copper nitrate, sodium hydroxide, etc.
|
Thanks. The section was revised and the producer of all used materials: acetone, copper nitrate, sodium hydroxide etc, were added
|
|
Extraction procedure is ”4 g of powder into 150 ml of deionized water, followed by 20 min of stirring and boiling“. Please explain why you used this procedure and this ratio of solid/liquid. Also, having in mind the boiling step, it is possible that most of the significant bioactive compounds are destroyed due to degradation. Therefore, it is necessary to optimize this extraction procedure and chemically characterize obtained extracts.
|
Thanks. The procedure and ratios gave results for the study of (Sumitha et al., 2016), hence I also used it.
|
|
It is stated that the synthesis of copper oxide nanoparticles was conducted according to reference 62. Since it is well-known that some parameters such as extract volume, concentration, time, pp, mm, temperature, pH value, temperature, etc. can have an influence on the synthesis of nanoparticles, my recommendation is to include an explanation of influence of some parameters and explain why you chose the specific parameters. Also, pH value should be included. the producer of all used materials: acetone, copper nitrate, sodium hydroxide, etc.
|
Thanks for your brilliant suggestion. We are currently busy with optimization studies in some of our future studies not this present work. Our target in this study was physiological pH so that we can use the material for electrochemical detection of some biological molecules. |
|
Regarding antibacterial activity, it would be an improvement if you include an antibiotic as a reference.
|
Thanks for the recommendation. Due to Covid-19 restrictions and closing of the University I cannot carry out more experiments.
|
Reviewer 3 Report
The article “Spectroscopic and antibacterial properties of CuONPs from orange, lemon and tangerine peel extracts: potential for combating bacterial resistance” describes the synthesis and characterization of CuO nanoparticles obtained with some peel extracts from citrus.
First of all, the English style and spelling should receive a major polishing. The spelling mistakes are too many to be mentioned.
Abstract should be reorganized without headlines background, methods results etc. Every time abstract should contains the most important information like most important findings and results. Some values are needed.
Transmission electron microscopy (TEM), Scanning Electron Microscopy (SEM) – must be written in the same way. UV-Vis and UV-vis? Copper Nitrate Cu(NO3)2 powder, and Sodium hydroxide (NaOH)..... 1 M Sodium Hydroxide; NPs and Nps. And such examples are to be found across the manuscript.
Authors should respect some conventions: American type culture collection has each word with capital letter; UV-Vis, mL;
Keywords: "nanoparticles,, , Ultraviolet-visible spectroscopy; Energy-dispersive X-ray spectroscopy (EDS), Fourier transform infrared spectrophotometer (FTIR)" – extra comas and characterization methods are not suitable for this section.
References [7,8,9,10] must be condensed [7-10].
Introduction part should be improved with more pertinent statements. “Nanoparticles are tiny solid materials with a structural length of a few “nanometres” (spelling). They are small... “. The definition for nano is to have at least one dimension under 100 nm. Nanoparticles therefore should have a diameter smaller than 100 nm. Nowadays indeed the aim is to obtain smaller and smaller nanoparticles, so I understand the statement with few nanometers, but authors should aim to improve such simple statements.
Row 122: “Signifies an increase in the number of infectious diseases, indicating a major constrain facing the health industry.” – Who signifies?
Row 134 “From this study...” I think should be “For this study...”
The interpretation in UV-Vis section is confuse. The peaks are found at ~ 290 nm, which is ok, but authors are saying “These findings adhere to previous reports in which the UV-vis spectra for copper oxide nanoparticles ranged from 300 nm-1000 nm”. So they are outside the range. Moreover is unknown what are they trying to explain with such broad range that is UV-Vis-NIR.
Row 152 “In contrast;” why semicolon? Plural of spectrum is spectra not spectrums like in row 153.
SEM images should have a legend with scale/magnification. SEM images are rather poor quality. Authors should try to put better ones.
In bibliography section there are different spaces between lines. Some references do not have same style, like bold year. Some years have letters.
Author Response
Response to Reviewer 3
Title: Spectroscopic and antibacterial properties of CuONPs from orange, lemon and tangerine peel extracts: potential for combating bacterial resistance
|
Comments |
Response
|
|
The article “Spectroscopic and antibacterial properties of CuONPs from orange, lemon and tangerine peel extracts: potential for combating bacterial resistance” describes the synthesis and characterization of CuO nanoparticles obtained with some peel extracts from citrus. First of all, the English style and spelling should receive a major polishing. The spelling mistakes are too many to be mentioned. Abstract should be reorganized without headlines background, methods results etc. Every time abstract should contains the most important information like most important findings and results. Some values are needed.
|
Thanks for the comment. The whole manuscript has been checked for spelling and grammatical errors. The abstract was rephrased, and some of the information was removed, while important information retained.
|
|
Transmission electron microscopy (TEM), Scanning Electron Microscopy (SEM) – must be written in the same way. UV-Vis and UV-vis? Copper Nitrate Cu(NO3)2 powder, and Sodium hydroxide (NaOH)...1 M Sodium Hydroxide; NPs and Nps. And such examples are to be found across the manuscript. Authors should respect some conventions: American type culture collection has each word with capital letter; UV-Vis, mL.
|
Thanks, they were all corrected |
|
Keywords: "nanoparticles,, , Ultraviolet-visible spectroscopy; Energy-dispersive X-ray spectroscopy (EDS), Fourier transform infrared spectrophotometer (FTIR)" – extra comas and characterization methods are not suitable for this section.
|
Thanks for your comment. Keywords session revised accordingly.
|
|
References [7,8,9,10] must be condensed [7-10]
|
Thanks, it was condensed to [7-10] on line 46
|
|
Introduction part should be improved with more pertinent statements. “Nanoparticles are tiny solid materials with a structural length of a few “nanometres” (spelling). They are small... “. The definition for nano is to have at least one dimension under 100 nm. Nanoparticles therefore should have a diameter smaller than 100 nm. Nowadays indeed the aim is to obtain smaller and smaller nanoparticles, so I understand the statement with few nanometers, but authors should aim to improve such simple statements. Row 122: “Signifies an increase in the number of infectious diseases, indicating a major constrain facing the health industry.” – Who signifies?
|
Thanks, I mentioned that it is signified by the World Health Organization
|
|
Row 134 “From this study...” I think should be “For this study...”
|
Thanks, it was corrected to “For this study” |
|
The interpretation in UV-Vis section is confuse. The peaks are found at ~ 290 nm, which is ok, but authors are saying “These findings adhere to previous reports in which the UV-vis spectra for copper oxide nanoparticles ranged from 300 nm-1000 nm”. So they are outside the range. Moreover is unknown what are they trying to explain with such broad range that is UV-Vis-NIR.
|
Thanks for the comment, there was a mistake that was fixed in the range
|
|
Row 152 “In contrast;” why semicolon? Plural of spectrum is spectra not spectrums like in row 153.
|
Thanks, the semicolon was removed in row 152, and the plural of spectrum was corrected
|
|
SEM images should have a legend with scale/magnification. SEM images are rather poor quality. Authors should try to put better ones.
|
Thank you very much, better ones were added.
|
|
In bibliography section there are different spaces between lines. Some references do not have same style, like bold year. Some years have letters
|
Thanks, they all have the same style now |
Round 2
Reviewer 2 Report
The authors improved the manuscript according to comments, therefore, I recommend accepting the work.
Reviewer 3 Report
Although the authors have addressed some of the problems identified, they should try also solve the problem with SEM images, (fig 5 and 6). There is no scale on the images. One can look at them and decide that each rod has 2 cm length for example. Please discuss the problem with the SEM technician to provide you at least magnification order, which can be written down in the figure's legend.